# Experience-dependent serotonergic signaling in glia regulates targeted synapse elimination

Vanessa Kay Miller [ID] [1], Kendal Broadie [ID] [1,2,3,4] *

1 Department of Biological Sciences, Vanderbilt University and Medical Center, Nashville, Tennessee, United States of America, 2 Department of Cell and Developmental Biology, Vanderbilt University and Medical Center, Nashville, Tennessee, United States of America, 3 Kennedy Center for Research on Human Development, Vanderbilt University and Medical Center, Nashville, Tennessee, United States of America, 4 Vanderbilt Brain Institute, Vanderbilt University and Medical Center, Nashville, Tennessee, United States of America

* kendal.broadie@vanderbilt.edu

**Data Availability Statement:** The Raw Data quantifications are provided in S1 Data. Raw images will be available in Harvard Dataverse under the "Kendal Broadie Dataverse". The dataset that corresponds to this article will be titled "Miller and

## Abstract

The optimization of brain circuit connectivity based on initial environmental input occurs during critical periods characterized by sensory experience-dependent, temporally restricted, and transiently reversible synapse elimination. This precise, targeted synaptic pruning mechanism is mediated by glial phagocytosis. Serotonin signaling has prominent, foundational roles in the brain, but functions in glia, or in experience-dependent brain circuit synaptic connectivity remodeling, have been relatively unknown. Here, we discover that serotonergic signaling between glia is essential for olfactory experience-dependent synaptic glomerulus pruning restricted to a well-defined *Drosophila* critical period. We find that experience-dependent serotonin signaling is restricted to the critical period, with both (1) serotonin production and (2) 5-HT$_{2A}$ receptors specifically in glia, but not neurons, absolutely required for targeted synaptic glomerulus pruning. We discover that glial 5-HT$_{2A}$ receptor signaling limits the experience-dependent synaptic connectivity pruning in the critical period and that conditional reexpression of 5-HT$_{2A}$ receptors within adult glia reestablishes "critical period-like" experience-dependent synaptic glomerulus pruning at maturity. These results reveal an essential requirement for glial serotonergic signaling mediated by 5-HT$_{2A}$ receptors for experience-dependent synapse elimination.

## Introduction

The brain first receives information from the environment during early critical periods and uses this input to optimize neural circuitry via large-scale changes in synapse connectivity [1,2]. Critical periods are defined as opening with sensory experience onset, a transiently reversible and dramatically heightened synaptic remodeling capacity, and then permanent closure resulting in the consolidation of mature brain circuits [3,4]. The closing of critical period remodeling limits later behavioral adaptability and prevents correction of subsequent impairments from injury, trauma, or disease but is presumed to be necessary to secure the maintained stability of brain circuit connectivity [5,6]. The large-scale changes in brain circuitry

Broadie PLOS Biology 2024". Link: https://dataverse.harvard.edu/dataverse/kendalbroadie#.

**Funding:** K.B. received support from National Institute of Health (grant nr.: NS132867) https://reporter.nih.gov/project-details/10878362. The funders had no role in study design, data collection and analysis, decision to publish, or preparation of the manuscript.

**Competing interests:** The authors have declared that no competing interests exist.

**Abbreviations:** AL, antennal lobe; ASD, autism spectrum disorder; CSDn, contralaterally projecting, serotonin-immunoreactive deutocerebral neurons; dpe, days post-eclosion; DSHB, Developmental Studies Hybridoma Bank; EB, ethyl butyrate; FXS, Fragile X syndrome; ID, intellectual disability; LTD, long-term depression; mGluR, metabotropic glutamate receptor; OD, ocular dominance; OE, overexpression; OSN, olfactory sensory neuron; PBS, phosphate buffered saline; PFA, paraformaldehyde; PTSD, posttraumatic stress disorder; RNAi, RNA interference; RT, room temperature; SSRI, selective serotonin reuptake inhibitor; Trhn, tryptophan hydroxylase.

during critical periods occur through dynamic fluctuations between 2 opposing remodeling processes: synapse formation and synapse elimination. Both the genesis and pruning of synapses is tightly regulated by glia [7,8], with critical period remodeling overall characterized by the large net loss of synapses directly mediated by experience-targeted glial phagocytosis [9–12]. Precise glial pruning is important to properly streamline information flow by adapting brain circuit synaptic connectivity to the unpredictable demands of a highly variable environment [13,14]. In mammals, microglia and astrocytes function as the phagocytes for synapse elimination, with multiple signaling cues to target and prune away unwanted synapses [14–16]. Microglia are the innate immune cells of the brain, and astrocytes are closely associated with synapses. Both glial classes can function as either primary or secondary phagocytes, with orchestrated roles in the engulfment and removal of neuron cell bodies, proximal dendritic arbors, and distal axonal synapses [17,18]. Microglia are key synaptic phagocytes, but astrocyte glia are reportedly the phagocytes mediating the elimination of excitatory synapses during the experience-dependent pruning of synaptic connections in adult mice [15,19,20]. In contrast, the glial mechanisms mediating synaptic pruning during experience-dependent critical period brain circuit remodeling have been much less studied. In particular, we know little about the molecular signaling mechanisms underlying critical period glial function.

Serotonin (5-HT) signaling plays requisite foundational roles mediating brain plasticity [21,22]. Serotonergic cells uniquely express tryptophan hydroxylase (Trhn), the rate-limiting enzyme for serotonin biosynthesis [23]. Serotonin signaling regulates both excitatory and inhibitory synapses, functioning in a gatekeeping mechanism controlling synaptic output and ratio changes [24,25]. Downstream, the G-protein-coupled 5-HT$_{2A}$ receptor (5-HT$_{2A}$R) regulates plasticity signaling [26,27] and is expressed in neurons and glia, including microglia and astrocytes [28–31]. 5-HT$_{2A}$R has long been closely linked to learning and memory [32], polarizing synaptic modifications that drive both long-term depression (LTD) and potentiation [33]. Importantly, the 5-HT$_{2A}$R has emerging roles in regulating brain circuit maturation and remodeling [21,34]. Serotonin signaling defects are linked to numerous neurological disorders, such as autism spectrum disorder (ASD), posttraumatic stress disorder (PTSD), depression, and schizophrenia [35,36], with serotonergic drugs at the forefront of patient symptom management [37,38]. Moreover, other serotonin pathway drugs (e.g., LSD, psilocybin) appear to increase the capacity in the adult brain for circuit remodeling [39,40], with the novel aspirational objective of reopening "critical period-like" remodeling in adults widely touted as a panacea for a myriad of mature brain limitations and impairments [41–43]. However, the putative role of serotonin signaling in experience-dependent brain circuit remodeling still remains largely unknown. There is no connection linking serotonergic signaling to glial function in synapse pruning during the early-life critical period, let alone such a role in later reopening this remodeling capacity in adults. Here, we employ a well-characterized critical period in the *Drosophila* genetic model system to test the requirements for serotonin signaling in sensory experience-dependent glial synapse pruning in both the juvenile and mature adult brain. We discover experience-dependent serotonergic signaling within glia is essential for targeted synapse pruning during the early-life critical period and that the conditional introduction of serotonin signaling in adult glia reopens this experience-dependent synapse pruning mechanism at maturity.

## Results

### Experience-dependent critical period serotonin signaling and synaptic glomerulus pruning

The *Drosophila* genetic model has a precisely defined early-life critical period [44,45], well-characterized glial phagocytes [46,47], and a sophisticated transgenic toolkit for cell-targeted

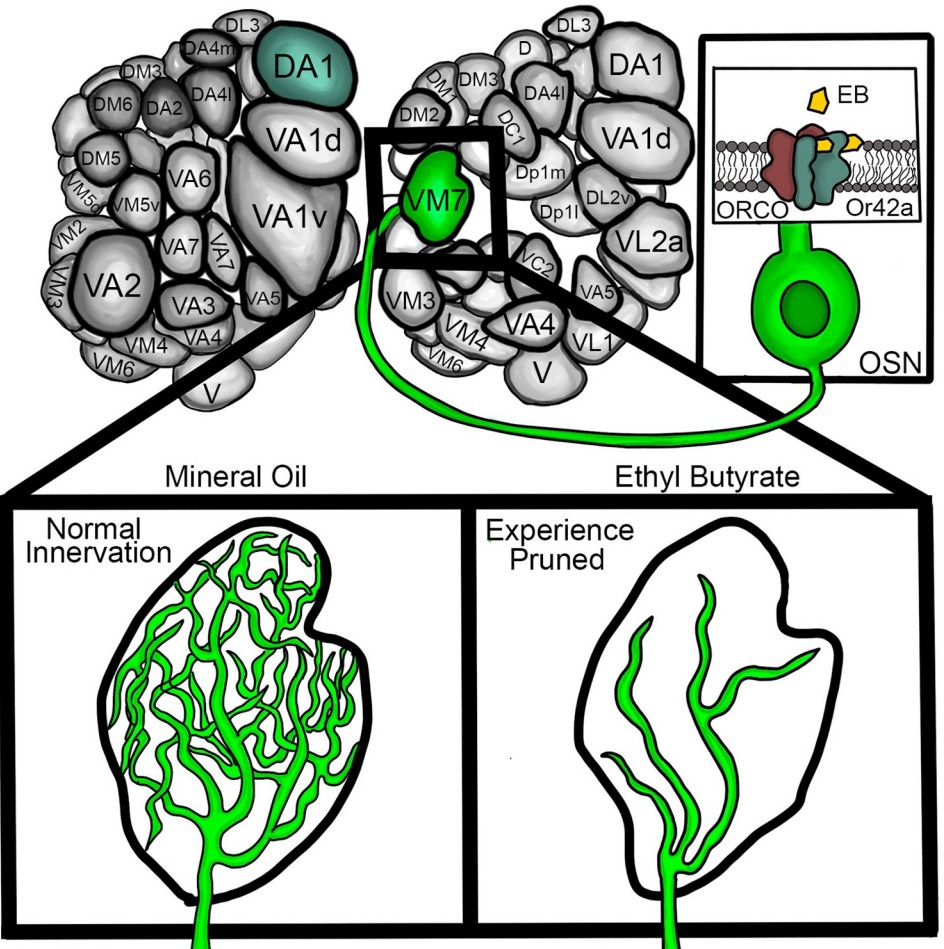

**Fig 1. Mapped *Drosophila* juvenile brain AL synaptic glomeruli exhibit olfactory experience-dependent pruning.**
Specific OSN classes innervate mapped synaptic glomeruli in the 2 brain hemispheres of the AL (top). Or42a OSNs
project specifically to the VM7 glomerulus (highlighted in green). The Or42a receptor complexed to the essential
ORCO subunit bind EB odorant to trigger an olfactory response (box on the right). Other mapped glomeruli, such as
DA1 (blue), do not respond to EB odorant. In the juvenile critical period, EB olfactory experience selectively prunes
Or42a OSN innervation of the VM7 synaptic glomerulus (bottom). The odorant vehicle mineral oil control exhibits
normal innervation (left), whereas transient exposure to EB odorant during the critical period result in experience-
dependent pruning (right). The glial pruning process is odorant dose-dependent and temporally restricted to the
critical period. AL, antennal lobe; EB, ethyl butyrate; OSN, olfactory sensory neuron; VM7, ventromedial 7.

manipulation of serotonergic signaling [24,25]. In the *Drosophila* juvenile brain, the
completely mapped synaptic glomeruli within the antennal lobe (AL) [48,49] are each inner-
vated by defined olfactory sensory neuron (OSN) synaptic terminals (Figs 1 and 2A, top left).
Single-receptor class OSNs respond to a specific odorant synapse onto the projection neurons
within each glomerulus [50]. The ethyl butyrate (EB)-responsive Or42a receptor OSNs inner-
vate the VM7 synaptic glomeruli within each hemisphere (Figs 1 and 2A, bottom left). Or42a
receptor-driven expression of the membrane marker mCD8::GFP is used to specifically label
this synaptic innervation [51]. Serotonin (5-HT) signaling is known to regulate OSN synaptic
connectivity (Fig 2A, top right) [52]. Labeling with a well-characterized anti-serotonin (5-HT)
antibody reveals serotonergic puncta throughout the AL synaptic neuropil. Glia are abundant
surrounding this neuropil (Fig 2A, bottom right) and extend projections into the synaptic glo-
meruli [11]. Glia act as phagocytes to prune the synaptic glomeruli and thereby remodel

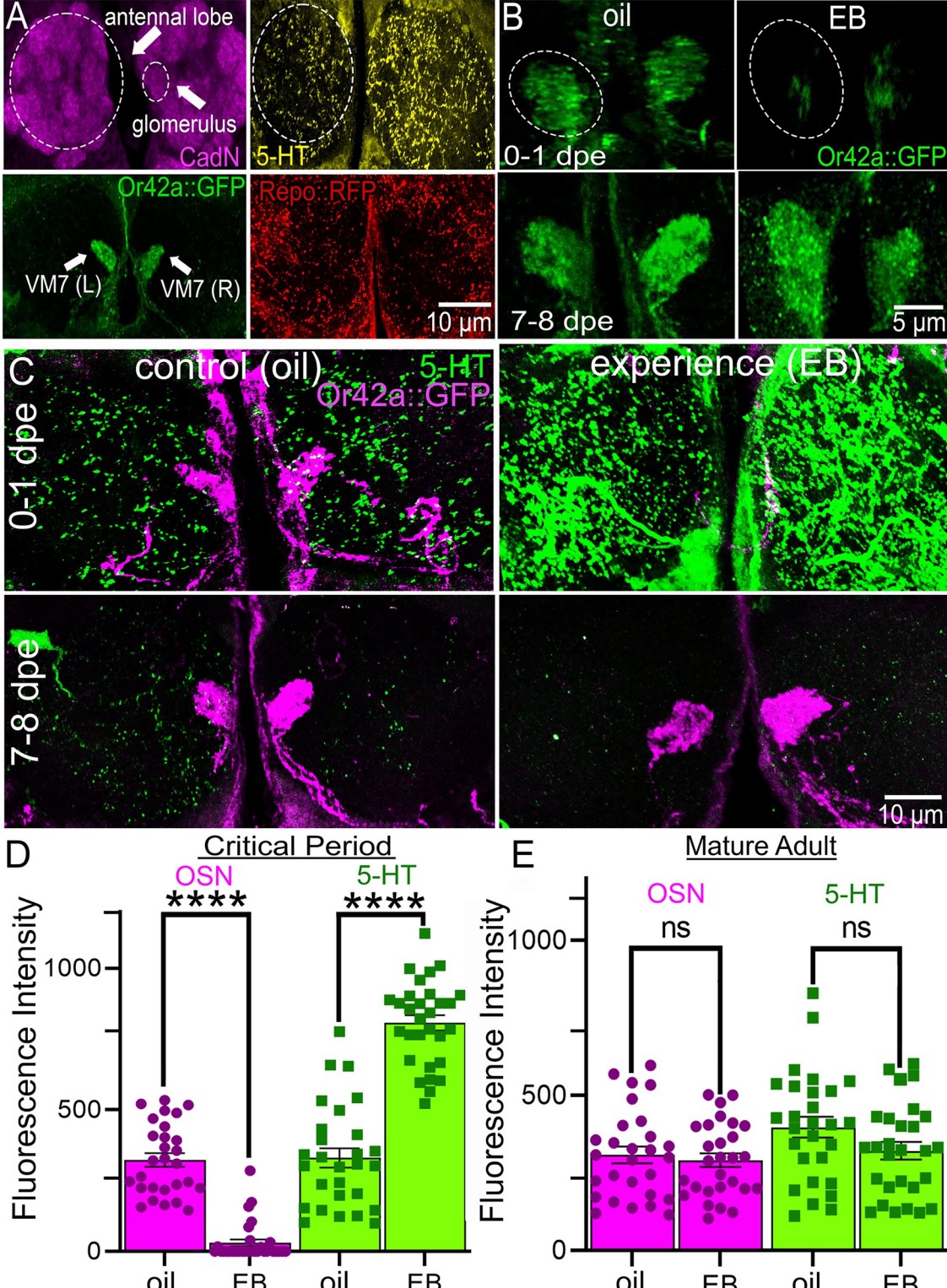

**Fig 2. Experience-dependent and temporally restricted serotonin signaling and synaptic glomeruli pruning.** (**A**) *Drosophila* brain AL synaptic glomeruli labeled with antibodies for CadN (magenta; top left). Paired VM7 glomeruli innervation labeled with Or42a odorant receptor-driven mCD8::GFP (Or42::GFP, green; bottom left). The same neuropil labeled with an antibody for serotonin (5-HT, yellow; top right), and glia labeled with glial *repo*-Gal4-driven UAS-mCD8::RFP (Repo>RFP, red; bottom right). (**B**) Timed exposure to either the odorant vehicle (oil, left) or 25% EB in mineral oil (right) for 24 hours from 0–1 dpe (critical period; top) or

7–8 dpe (adult maturity, bottom). Or42a receptor-driven mCD8::GFP (Or42::GFP, green) membrane labeling of the OSN innervation of the VM7 glomeruli (dashed circles) shows experience-dependent pruning only in the 0–1 dpe critical period, with none in 7–8 dpe adults. (**C**) Serotonin (5-HT, green) is strongly up-regulated by EB experience during the experience-dependent pruning of the Or42a OSN synaptic glomeruli (Or42::GFP, magenta) in the 0–1 dpe critical period, but there is no EB experience-dependent serotonin signaling or pruning in the 7–8 dpe adults. (**D**) Quantification of the Or42a neuron innervation (OSN, magenta) and serotonin (5-HT, green) fluorescence intensity within the VM7 glomerulus in the oil vehicle control and EB experience conditions in the critical period (0–1 dpe). Two-way ANOVA with Tukey's multiple comparison shows a highly significant decrease in the Or42a OSN innervation ($n = 28$/condition, $p = 7.45 \times 10^{-11}$) and increase in serotonin ($n = 28$/condition, $p = 4.10 \times 10^{-14}$). (**E**) The same quantification in mature adults (7–8 dpe) shows no experience-dependent change in either the Or42a OSN innervation ($n = 28$/condition, $p = 0.9738$) or serotonin signaling ($n = 28$/condition, $p = 0.9965$). All the individual data points are shown with mean ± SEM. Significance is indicated as $p < 0.0001$ (****) and $p > 0.05$ (not significant, ns). Source data can be found in S1 Data. AL, antennal lobe; CadN, N-Cadherin; dpe, days post-eclosion; EB, ethyl butyrate; OSN, olfactory sensory neuron; VM7, ventromedial 7.

innervation connectivity [11,53]. Or42a receptor OSNs have a precisely timed, early-life critical period, which is tightly temporally restricted, transiently reversible, and odorant dose-dependent (Fig 2B) [44,54]. In response to timed 24-hour EB odorant experience, the Or42a OSN synaptic glomeruli innervation is strongly pruned compared to the vehicle alone control (oil), with pruning only in the juvenile brain critical period (0 to 1 days post-eclosion (dpe); Fig 2B, top). In contrast, there is no significant VM7 synaptic glomeruli innervation pruning in response to the exact same EB odorant sensory experience in mature adults (7 to 8 dpe, Fig 2B, bottom). Serotonin signaling is well established to modulate the olfactory synaptic connectivity in the adult brain via characterized serotonergic neuron innervation [24,25,52]; however, nothing is known about serotonin signaling roles within the juvenile brain critical period.

We discover serotonin signaling is dramatically up-regulated in response to critical period EB odorant experience, coincident with temporally restricted VM7 synaptic glomeruli pruning, but the same experience in mature adults results in no significant signaling or pruning (Fig 2C). Both synaptic glomeruli innervation and serotonin labeling is done in colabeled brains in response to timed EB experience trials at the 2 time points. In the critical period, 24-hour EB odorant experience from 0 to 1 dpe strikingly up-regulates serotonin (5-HT, green) compared to the vehicle control (oil), while also driving EB-responsive Or42a neuron synaptic glomeruli pruning (magenta) compared to the vehicle control (Fig 2C, top). In sharp contrast, identical EB experience in mature adults (7 to 8 dpe) causes no detectable change in either serotonin signaling or VM7 innervation (Fig 2C, bottom). Fluorescence quantification of the timed oil odorant vehicle control compared to the EB experience condition reveals the highly significant synaptic glomeruli pruning (magenta) during the 0- to 1-dpe critical period (Fig 2D, left). Likewise, anti-serotonin (5-HT, green) fluorescence quantification shows coincident highly significant experience-dependent serotonin signaling up-regulation during the same restricted 0- to 1-dpe critical period (Fig 2D, right). Conversely, 24-hour EB experience in the mature adults from 7 to 8 dpe results in absolutely no change in the OSN synaptic innervation (magenta) compared to the matched oil vehicle control (Fig 2E, left). Consistently, the same EB experience in mature adults causes no detectable change in serotonin signaling (green) compared to control (Fig 2E, right). These results demonstrate that experience-dependent serotonin signaling and synaptic glomeruli pruning are both temporally restricted to the critical period. Consistent with an OSN-specific sensory input mechanism, critical period EB experience genetically restricted to Or42a neurons alone induces both serotonin signaling and synaptic glomeruli pruning (S1A and S1B Fig). To test OSN-specific serotonin regulation, a non-EB-responsive glomerulus (DA1; Fig 1) was imaged for experience-dependent serotonin changes. In response to EB exposure in the critical period, DA1 exhibits no 5-HT up-regulation (S1C and S1D Fig) [55]. Likewise, an independent brain region (optic lobe) also shows no 5-HT up-regulation in response to critical period EB exposure (S1E and S1F Fig). These

findings indicate that experience-dependent serotonin signaling in the critical period is circuit-localized in the juvenile brain. We next turned to testing the possible role for critical period experience-dependent serotonin signaling in synaptic glomeruli pruning by assaying the cellular requirement for serotonin.

## Serotonin signaling by glia (but not neurons) is essential for synaptic glomeruli pruning

To begin to test a possible requirement for experience-dependent serotonergic signaling in the experience-dependent synaptic glomeruli pruning during the critical period, we first turned to using cell-targeted RNA interference (RNAi) against Trhn, an essential enzyme for serotonin synthesis [23,56]. Our obvious primary candidate was serotonergic signaling from neurons, so we first employed the pan-neuronal *elav*-Gal4 driver [57] to express UAS-*Trhn* RNAi and thus block serotonin production in all neurons [58]. To further refine this analysis, we next then eliminated serotonin signaling specifically from the contralaterally projecting, serotonin-immunoreactive deutocerebral neurons (CSDn), which are the well-characterized serotonergic neurons that innervate the AL synaptic glomeruli [24]. We fully expected all serotonergic signaling to come from only these CSDn neurons. However, glia have also been reported to participate in serotonergic signaling pathways [59,60], glial phagocytosis is well established to mediate synaptic glomeruli pruning during critical periods [61,62]; and glia are well mapped within the *Drosophila* brain olfactory circuit (Fig 2A) [44,47]. Therefore, for a complete analysis, we also employed the glia-specific *repo*-Gal4 to drive UAS-*Trhn* RNAi in glia. Glia have never been reported to mediate serotonergic signaling, so we tested Trhn and 5-HT labeling in glia, and also experience-dependent glial serotonergic signaling (S2 Fig). In the UAS-*Trhn* RNAi control (no Gal4 driver) and all 3 cell-targeted knockdowns, we paired 24-hour critical period exposure (0 to 1 dpe) of the odorant vehicle control (oil) versus 25% EB odorant (% v/v EB in oil). We tested for experience-dependent pruning of Or42a OSN innervation in the VM7 glomerulus, with all synaptic glomeruli labeled with an N-Cadherin antibody (grey scale), and VM7 innervation labeled by Or42a receptor-driven membrane mCD8::GFP in an intensity heat-map (color scale). Representative images and quantification of the 3-dimensional Or42a OSN innervation volume within the VM7 synaptic glomeruli are shown in Fig 3.

In the UAS-*Trhn* RNAi transgenic control, 24-hour EB experience (0 to 1 dpe) drives strong VM7 synaptic glomeruli pruning compared to the oil vehicle (Fig 3A, top). The vehicle control (left) shows intense fluorescence occupying the entire VM7 glomerulus, whereas EB experience causes a striking loss of labeling intensity and volume (right). To our surprise, blocking serotonin synthesis in glia (*repo*-Gal4-driven UAS-*Trhn* RNAi) prevents this experience-dependent pruning (Fig 3A, middle). Vehicle control (left) and EB experience (right) innervation is indistinguishable in the absence of serotonin signaling from glia. Glial serotonin synthesis in the critical period is confirmed by colabeling for glial nuclei (anti-Repo), serotonin (anti-5-HT), and *Trhn*-Gal4 driving a cell membrane marker (UAS-mCD8::GFP; S2A and S2B Fig). Critical period EB experience induces serotonin production in these glia (S2A–S2C Fig), with an experience-dependent increase of both the Trhn marker and 5-HT colocalization with these same glia (S2D and S2E Fig). Consistently, EB experience-dependent serotonin signaling is blocked by *repo*-Gal4-driven UAS-*Trhn* RNAi in glia (S3 Fig; see below). In contrast, blocking serotonin production in neurons has no effect on experience-dependent synaptic glomeruli pruning (Fig 3A, bottom). Compared to the oil vehicle (left), the Or42a neuron innervation is strongly pruned by critical period EB exposure (right), to the same degree as in the control. Consistently, *elav*-Gal4-driven UAS-*Trhn* RNAi blocks serotonin production in CSDn soma but has a weaker effect on experience-dependent serotonin signaling in the VM7 glomerulus

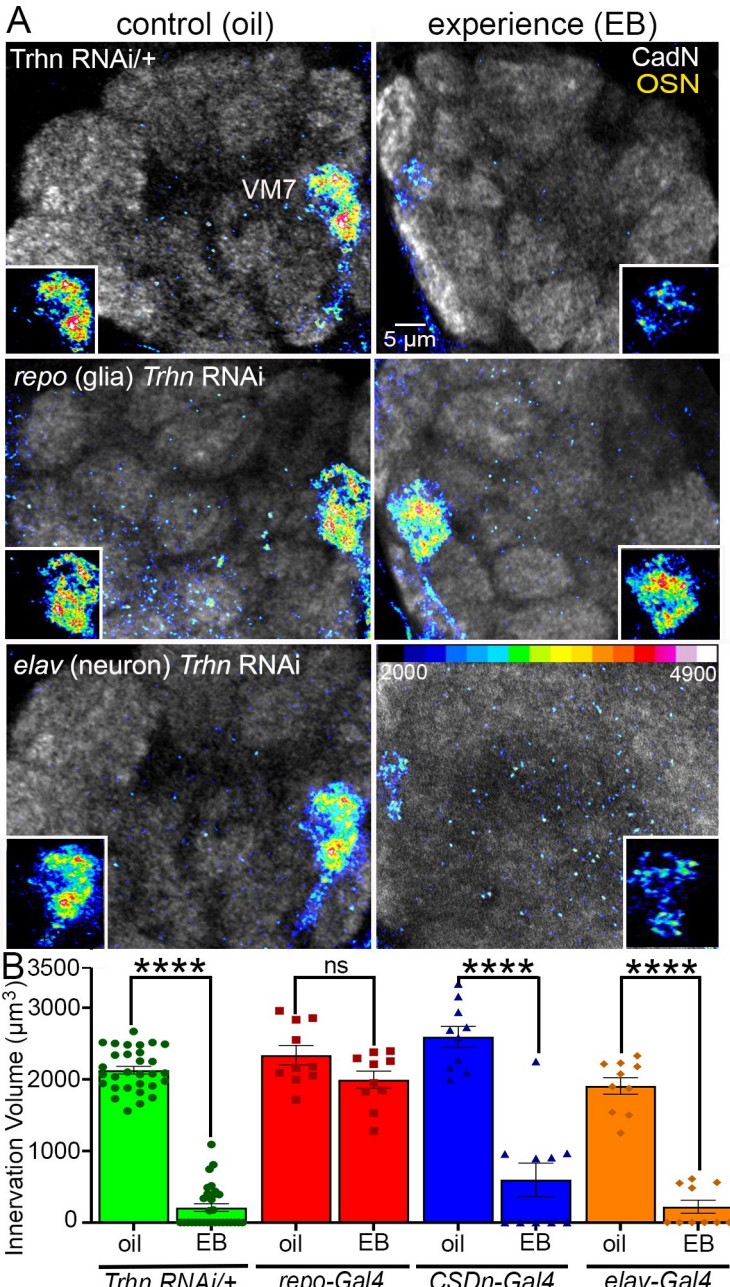

**Fig 3. Glial serotonin signaling is required for experience-dependent pruning of synaptic glomeruli in the critical period.** (**A**) CadN (greyscale) labeling of synaptic glomeruli with Or42a OSN innervation of the VM7 glomerulus shown as a colored heat-map (16 LU scale, bottom right). The insets show single-channel OSN images in VM7. The top row is the *Trhn* RNAi alone transgenic control (control: *w^{1118}; Or42a-mCD8::GFP/+; UAS-Trhn RNAi/+*) with the oil vehicle control (left) and 25% EB experience (right) for 24 hours from 0–1 dpe in the critical period. Second row is glia-specific *repo*-Gal4-driven *Trhn* RNAi (*repo* (glia): *w^{1118}; Or42a-mCD8::GFP/+; UAS-Trhn RNAi/repo-Gal4*) showing a total block of experience-dependent synaptic glomeruli pruning. Bottom row is pan-neuronal *elav*-Gal4-driven *Trhn* RNAi (*elav* (neuron): *w^{1118}; Or42a-mCD8::GFP/elav-Gal4; UAS-Trhn RNAi/+*) showing no effect on the EB experience-dependent synaptic glomeruli pruning. (**B**) Quantification of the Or42a OSN innervation volume in the undriven *Trhn* RNAi control, glial-targeted *Trhn* RNAi, and 2 neuron-targeted *Trhn* RNAi lines; *TRHN* RNAi control (green, left), glial RNAi (red, second from left), serotonergic CSDn neuron RNAi (blue; *w^{1118}; Or42a-Gal4/+; UAS-Trhn RNAi/GMR60F02-Gal4*), and the pan-neuronal *elav* RNAi (orange, right). Two-way ANOVA with Tukey's multiple comparison shows significant glial pruning of the Or42a OSN innervation volume in the *Trhn* RNAi control ($n = 30$/condition, $p = 4.00 \times 10^{-14}$), as well as both neuronal *Trhn* RNAi conditions (serotonergic *CSDn*-Gal4: $n = 10$/condition, $p = 6.4 \times 10^{-14}$; pan-neuronal *elav*-Gal4: $n = 10$/condition, $p = 1.19 \times 10^{-14}$), but no significant synaptic

glomeruli pruning with the glial-targeted *Trhn* RNAi (*repo*-Gal4: $n = 10$, $p = 0.539$). All the individual data points are shown with the mean ± SEM. Significance is indicated as $p < 0.0001$ (****) and $p > 0.05$ (not significant, ns). Source data can be found in S1 Data. CadN, N-Cadherin; CSDn, contralaterally projecting, serotonin-immunoreactive deutocerebral neurons; dpe, days post-eclosion; EB, ethyl butyrate; OSN, olfactory sensory neuron; RNAi, RNA interference; *Trhn*, *tryptophan hydroxylase*; VM7, ventromedial 7.

compared to *repo*-Gal4-driven UAS-*Trhn* RNAi in glia (S3A–S3C Fig). Thus, glial *Trhn* knockdown impairs the EB experience-dependent 5-HT production in the VM7 synaptic glomerulus but not CSDn neurons, whereas neuronal *Trhn* knockdown effectively eliminates 5-HT from CSDn neurons (S3D and S3E Fig). Quantification of innervation volume shows highly significant pruning in the transgenic controls from EB experience compared to the oil vehicle (Fig 3B, left, green). In contrast, *repo*-Gal4-driven *Trhn* RNAi in glia results in no significant experience-dependent pruning (Fig 3B, red). Both oil vehicle and EB experience results in indistinguishable innervation compared to oil transgenic controls without EB exposure. The same quantification with serotonin synthesis blocked specifically in the serotonergic CSDn neurons with *GMR60F02*-Gal4 (Fig 3B, blue, $p = 6.4 \times 10^{-14}$), or in all neurons with *elav*-Gal4 (Fig 3B, orange, $p = 1.19 \times 10^{-14}$), shows normal synaptic glomeruli pruning. These surprising findings reveal that serotonin signaling from glia is essential for critical period experience-dependent synaptic glomeruli pruning, with no detectable involvement from serotonergic neurons. We next turned to testing for the cells responding to this experience-dependent glial serotonin signaling during the critical period.

## Glial 5-HT$_{2A}$ autoreceptors are necessary and rate-limiting for synaptic glomeruli pruning

Although serotonergic 5-HT$_{2A}$ G-protein-coupled receptors are well established to regulate brain circuit plasticity in neurons [32,39,40], little is known about possible developmental roles. To test functions in experience-dependent critical period circuit pruning, we first used *elav*-Gal4-driven *5-HT$_{2A}$R* RNAi in neurons [63], but found no detectable role for these receptors in neurons, with normally maintained experience-dependent synaptic glomeruli pruning compared to matched controls (S4A and S4B Fig). In addition to roles in neuronal synapses, 5-HT$_{2A}$ receptors are present in glial phagocytes, including microglia and astrocytes [29,59,60]. In neurons, 5-HT$_{2A}$R autocrine signaling is a well-established mechanism during synaptic regulation [64]. Together, this suggested a possible 5-HT$_{2A}$R self-signaling function in glia for experience-dependent critical period synaptic glomeruli pruning. To test this novel idea, we again used the glial-specific *repo*-Gal4 to drive UAS-*5-HT$_{2A}$R* RNAi and then assayed experience-driven VM7 synaptic glomeruli pruning. We find that 5-HT$_{2A}$ receptors within glia are required for EB experience-dependent critical period pruning (Fig 4A). In the *5-HT$_{2A}$R* RNAi transgenic control, 25% EB experience for 24 hours (0 to 1 dpe) again causes extensive Or42a neuron synaptic glomeruli pruning in comparison to the maintained VM7 innervation in the oil odorant vehicle condition (Fig 4A, top). In direct contrast, using glial-targeted *5-HT$_{2A}$R* RNAi with *repo*-Gal4 completely blocks experience-dependent synaptic glomeruli pruning, with both the vehicle control (oil) and EB experience conditions exhibiting indistinguishable levels of VM7 innervation (Fig 4A, bottom). Quantification of the Or42a OSN innervation volume shows critical period EB experience results in the significant glial pruning of VM7 synaptic glomeruli in the *5-HT$_{2A}$R* RNAi transgenic controls (Fig 4C, left, blue). Conversely, there is no significant innervation pruning with the glial-targeted *5-HT$_{2A}$R* RNAi (Fig 4C, red). We find 5-HT$_{2A}$R is elevated by EB experience only during the juvenile critical period (green), as mature adults show no 5-HT$_{2A}$R change in response to EB exposure

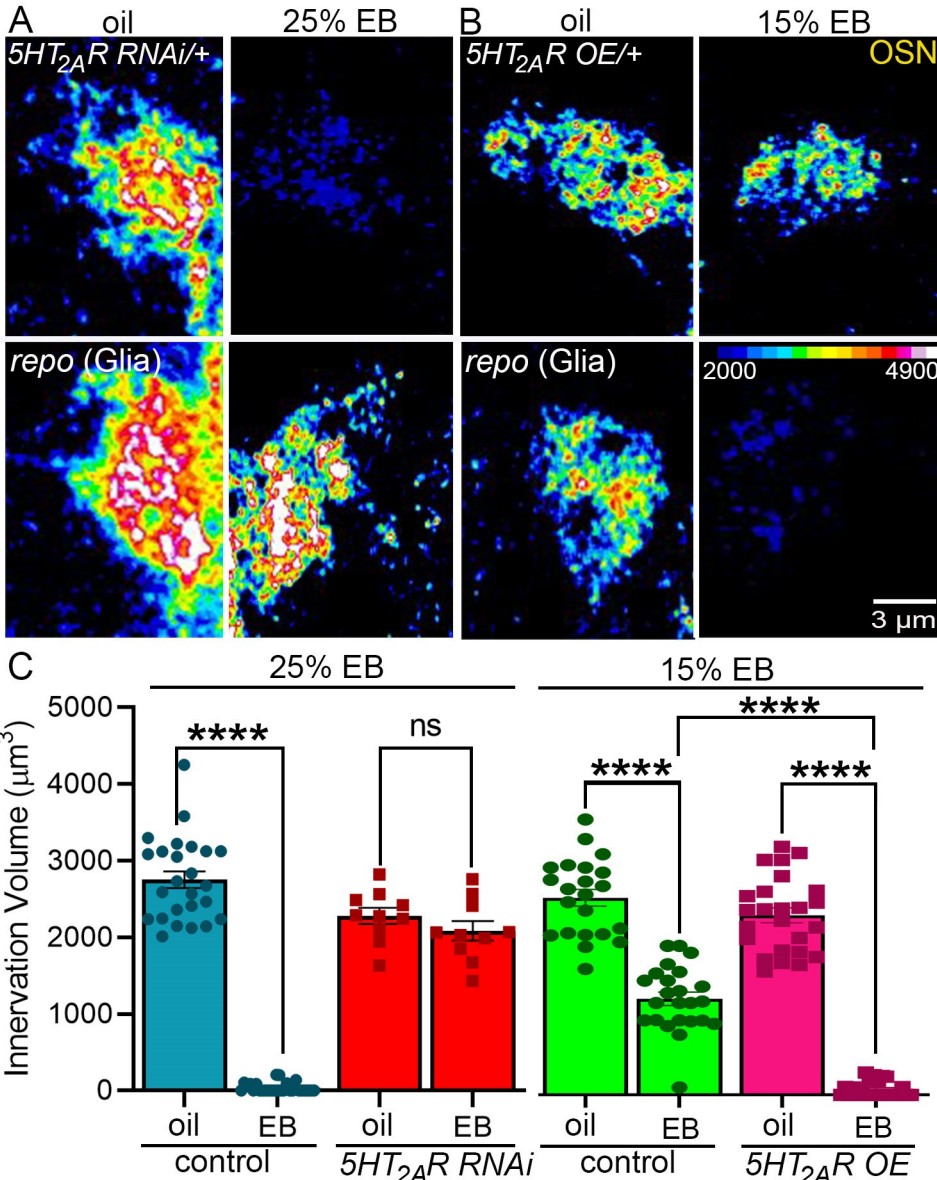

**Fig 4. Glial 5-HT$_{2a}$ receptors are necessary and rate limiting for experience-dependent synaptic glomeruli pruning. (A)** Or42a OSN innervation of VM7 glomeruli shown as a heat-map (color LU scale; lower right, in B). The top row is the 5-HT$_{2A}$ receptor RNAi control ($w^{1118}$; *Or42a-mCD8::GFP/+; UAS-5-HT$_{2A}$R RNAi/+*) with the odorant vehicle (oil, left) and 25% EB exposure (right) from 0–1 dpe showing the experience-dependent innervation pruning. The bottom row is the glial *repo*-Gal4-driven 5-HT$_{2A}$ receptor RNAi (*repo* (glia): $w^{1118}$; *Or42a-mCD8::GFP/+; UAS-5-HT$_{2A}$R RNAi/repo-Gal4*) showing a total blockade of innervation pruning. **(B)** The top row is the undriven 5-HT$_{2A}$ receptor OE control (5-HT$_{2A}$R OE/+: $w^{1118}$; *Or42a-mCD8::GFP/+; UAS-5-HT$_{2A}$R OE/+*) showing reduced synaptic glomerulus pruning with 15% EB from 0–1 dpe. Bottom row: glial *repo*-Gal4-driven 5-HT$_{2A}$R OE [*repo* (glia)]: $w^{1118}$; *Or42a-mCD8::GFP/+; UAS-5-HT$_{2A}$R OE/ repo-Gal4*) showing highly elevated pruning. **(C)** Quantification of the Or42a OSN innervation volume with 25% EB experience (left) in the transgenic RNAi control (blue) and glial *5-HT$_{2A}$ receptor* RNAi (red), and at 15% EB (right) in the *5-HT$_{2A}$R* OE control (green) and with *5-HT$_{2A}$R* OE (magenta). Two-way ANOVA with Tukey's multiple comparison shows highly significant glial pruning of the Or42a OSN innervation volume with the higher EB experience in the *5-HT$_{2A}$R* RNAi control ($n = 25$/condition, $p = 1.00 \times 10^{-15}$), but no significant pruning with glial-targeted *5-HT$_{2A}$R* RNAi ($n = 10$/condition, $p = 0.662$). Likewise, with 15% EB, there is significant glial pruning in the *5-HT$_{2A}$R* control ($n = 25$/condition, $p = 4.67 \times 10^{-10}$), but very much greater pruning with the glial-targeted *5-HT$_{2A}$R* OE ($n = 25$/condition, $p = 4.63 \times 10^{-10}$). The Or42a OSN innervation volume with glial *5-HT$_{2A}$R* OE is very significantly decreased in comparison to the control EB condition ($p = 4.64 \times 10^{-10}$). All the individual data points are shown with mean ± SEM. Significance is indicated as $p < 0.0001$ (****) and $p > 0.05$ (not significant, ns). Source data can be found in S1 Data. dpe, days post-eclosion; EB, ethyl butyrate; OE, overexpression; OSN, olfactory sensory neuron; RNAi, RNA interference; VM7, ventromedial 7.

(red; S5A and S5B Fig). Thus, we conclude glial 5-HT$_{2A}$ receptors are required for experience-dependent synaptic glomeruli pruning during the early-life critical period.

The above results show serotonergic 5-HT$_{2A}$ receptor signaling between glia is essential for glial pruning of synaptic glomeruli, but fail to test whether 5-HT$_{2A}$ receptors determine the extent of experience-dependent pruning. We therefore next tested whether 5-HT$_{2A}$ receptors within glia limit critical period synaptic glomeruli pruning. To test this possibility, we decreased the EB odorant concentration (15% EB v/v in mineral oil) to reduce the potency of the sensory experience and thereby decrease the extent of experience-dependent synaptic pruning. Under this new condition, we then tested whether glial *repo*-Gal4-driven UAS-*5-HT$_{2A}$R* overexpression (OE) would increase the extent of the experience-dependent synaptic glomeruli pruning (Fig 4B). In the *5-HT$_{2A}$R$^{OE}$*/+ transgenic controls, there is a dose-dependent decrease in the extent of synaptic pruning following 15% EB exposure compared to vehicle control (oil) for 24 hours from 0 to 1 dpe, but a reduced level of experience-dependent synaptic glomeruli pruning is apparent (Fig 4B, top). In comparison, *5-HT$_{2A}$R$^{OE}$* in glia results in an obvious increase in synaptic glomeruli pruning in the 15% EB condition (Fig 4B, bottom), to a degree indistinguishable from the higher 25% EB condition (compared to Fig 4A, top). Quantification of the *5-HT$_{2A}$R$^{OE}$*/+ transgenic controls shows reduced but still significant synaptic glomeruli pruning from the lower 15% EB critical period experience (Fig 4C, right, green). In comparison, glial-targeted 5-HT$_{2A}$ receptor OE significantly increases the extent of EB experience-dependent synaptic glomeruli pruning (Fig 4C, right, magenta). Neuronal *elav*-Gal4-driven *5-HT$_{2A}$R* RNAi strongly eliminates 5-HT$_{2A}$R expression in both control and EB exposure conditions, confirming knockdown specificity (S6A and S6C Fig). Importantly, glial *repo*-Gal4-driven *5-HT$_{2A}$R* RNAi and OE show the expected bidirectional changes in 5-HT$_{2A}$R levels, with *5-HT$_{2A}$R$^{OE}$* significantly elevating the EB experience-dependent 5-HT$_{2A}$R up-regulation (S6B and S6C Fig). Taken together, *5-HT$_{2A}$R* RNAi in glia completely blocks the synaptic glomeruli pruning with higher dose 25% EB exposure (Fig 4C, left), and *5-HT$_{2A}$R* OE in glia greatly increases pruning with the lower-dose 15% EB experience (right). This bidirectional glial 5-HT$_{2A}$ receptor regulation is highly significant in both conditions. Given that glial 5-HT$_{2A}$ receptors limit experience-dependent synaptic glomeruli pruning in the critical period, we next asked whether glial 5-HT$_{2A}$R OE could induce similar remodeling at maturity.

## Adult glial serotonergic signaling reopens experience-dependent synaptic glomeruli pruning

The lifelong debilitation from critical period associated impairments in juvenile brains has led to attempts to reopen "critical period-like" remodeling capacities in mature adults [41,42,65]. We found that 5-HT$_{2A}$ receptor OE in glia strongly increases experience-dependent synapse remodeling in the juvenile brain critical period, suggesting it might possibly also enable de novo experience-dependent remodeling in mature brains. To test this hypothesis, we employed the conditional, temperature-sensitive Gal80 (Gal80$^{ts}$) transcriptional repressor to reintroduce the 5-HT$_{2A}$ receptor only in adult glia and then assayed for resumption of experience-dependent synaptic glomeruli pruning at maturity. The Gal80$^{ts}$ repressor blocks Gal4-mediated transcription at a lower permissive temperature (18˚C) but is inactivated to allow Gal4 transcription at a higher restrictive temperature (28˚C) [66]. Wild-type adults show no detectable experience-dependent synaptic glomeruli pruning of the Or42a neuron innervation in VM7 glomeruli with EB odorant exposure in mature adults (Fig 2B and 2E). Likewise, transgenic Gal80$^{ts}$ adults in the permissive 18˚C temperature condition (blue), when the Gal80$^{ts}$ repressor remains active and 5-HT$_{2A}$ receptors are therefore not overexpressed in glia, also show no EB experience-dependent synaptic glomeruli pruning in either transgenic controls or

the repressed $5\text{-}HT_{2A}R$ OE condition (Fig 5A). EB experience in all of these adults does not detectably alter the VM7 innervation. Similarly, at the restrictive 28˚C temperature (red), when the Gal80$^{ts}$ repressor is inactive and $5\text{-}HT_{2A}$ receptors are overexpressed within the adult glia, the odorant vehicle control (oil) condition also shows no synaptic glomeruli pruning (Fig 5B, top). This is expected as there is no EB sensory experience to target VM7 synaptic connectivity remodeling in this condition. Quantification of 3-dimensional innervation volume in all 3 of these control conditions confirms there is no significant synaptic glomeruli pruning in mature adults at either the 18˚C or 28˚C temperatures in the absence of induced $5\text{-}HT_{2A}R$ OE in glia (Fig 5C, green and blue).

In direct contrast, conditional glial-targeted $5\text{-}HT_{2A}R$ OE in fully mature adults induces strong experience-dependent synaptic glomeruli pruning, which is indistinguishable from the critical period pruning mechanism (Fig 5B). At the restrictive 28˚C (red), the transgenic control lacking the $5\text{-}HT_{2A}R^{OE}$ construct shows no detectable change in the Or42a neuron innervation of the VM7 glomerulus following EB experience compared to the oil odorant vehicle alone (Fig 5B, left). Likewise, in Gal80$^{ts}$ animals at 28˚C with the repressor inactive and $5\text{-}HT_{2A}R^{OE}$ in glia, the odorant vehicle control (oil) similarly show no synaptic pruning, consistent with the lack of EB exposure experience (Fig 5B, right top). However, under these exact same conditions with EB experience, there is strong glial pruning of the Or42a neuron innervation of the VM7 glomerulus (Fig 5B, right bottom). Consistent with the synaptic glomeruli pruning, EB experience-dependent 5-HT up-regulation indistinguishable from the critical period response occurs exclusively in the repressor inactive condition (28˚C) when the $5\text{-}HT_{2A}R^{OE}$ construct is expressed (S7A and S7B Fig). This de novo experience-dependent synaptic glomeruli pruning is quite comparable in extent to the critical period pruning (compared to Figs 2–4), suggesting a full regeneration of the juvenile remodeling capacity. Quantification of the Or42a neuron innervation volume in the VM7 synaptic glomerulus for all 8 conditions is shown in Fig 5C. At the permissive 18˚C temperature (blue, left), when the Gal80$^{ts}$ repressor is active, the innervation volume shows a complete lack of pruning in the oil vehicle control versus the EB experience conditions (Fig 5C, left). No significant pruning occurs in the transgenic control (green, left) or in the repressed Gal80$^{ts}$ animals (blue, second from left). In the restrictive 28˚C temperature (red, right), when Gal80$^{ts}$ is inactive, the glial-targeted $5\text{-}HT_{2A}$ receptor expression drives highly significant experience-dependent pruning in mature adults (Fig 5C, right). No significant synaptic pruning happens in the transgenic control (green, left), but now significant EB experience-dependent synaptic glomeruli pruning occurs with the conditional $5\text{-}HT_{2A}$ receptor OE in adult glia (red, right). Similarly, 28˚C conditional glial *Trhn* OE (*Trhn*$^{OE}$) in mature adult glia also induces experience-dependent serotonin up-regulation and VM7 synaptic glomeruli pruning in response to EB experience (S8A–S8C Fig). These findings show that conditional adult glia-specific Trhn OE to drive serotonin production as well as serotonergic $5\text{-}HT_{2A}$ receptor OE targeted to adult glia both trigger the reopening of "critical period-like" synaptic glomerulus remodeling at maturity.

## Discussion

We discover glia-to-glia serotonin signaling as a novel mechanism of brain circuit remodeling via experience-dependent synaptic pruning in *Drosophila* [30,61,62]. Serotonergic signaling is well known to regulate juvenile brain circuit remodeling in mammals [67]. For example, interference with serotonergic signaling impairs visual cortex ocular dominance (OD) remodeling in the postnatal critical period [68], and serotonin modulates synapse maturation in the developing prefrontal cortex [69]. However, such studies underemphasize, or entirely neglect, glial participation in these mechanisms. We propose that up-regulation of glial serotonin signaling

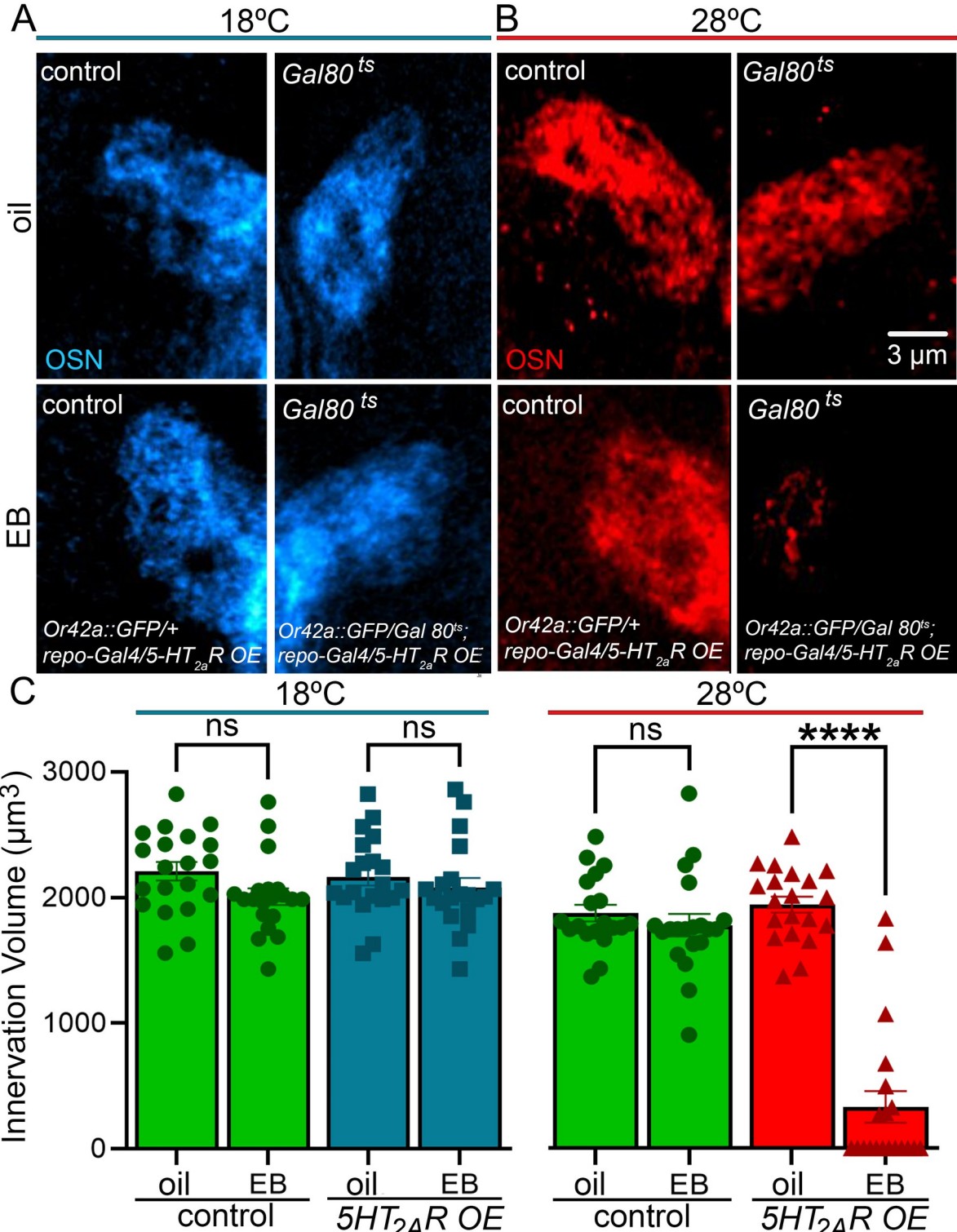

**Fig 5. Glia-targeted 5-HT$_{2A}$ receptor expression in adults reopens experience-dependent synaptic glomeruli pruning.** (**A**) Mature adult animals show no detectable experience-dependent pruning of the Or42a OSN innervation in VM7 glomeruli (blue) at 18˚C (Gal80$^{ts}$ permissive temperature). The top row shows the $w^{1118}$ genetic background control (control: $w^{1118}$; tubulin Gal80/Or42a-mCD8::GFP; repo-Gal4/+) and the Gal80$^{ts}$ transgenic control (*Gal80$^{ts}$*: $w^{1118}$; tubulin Gal80$^{ts}$/Or42a-mCD8:GFP; 5-HT$_{2A}$R OE/ repo-Gal4) with 24-hour mineral oil vehicle at 14–16 dpe (18˚C). The bottom row shows the same genotypes with 25% EB odorant experience for 24 hours at adult maturity (14–16 dpe, 18˚C). (**B**) At 28˚C (Gal80$^{ts}$ restrictive temperature; red), glial 5-HT$_{2A}$R OE enables experience-dependent synaptic

glomerulus pruning in mature adults. Top row is the 2 controls exposed to the oil vehicle only. Bottom row is the genetic background control ($w^{1118}$; tubulin Gal80$^{ts}$/Or42a-mCD8::GFP; repo-Gal4/+) and repressed Gal80$^{ts}$ condition ($w^{1118}$; tubulin Gal80$^{ts}$/Or42a-mCD8:GFP; 5-HT$_{2A}$R OE/ repo-Gal4) following 24-hour exposure to 25% EB experience in the mature adult. (**C**) Quantification of synaptic glomeruli innervation volumes at permissive 18°C (left) for control (green) and Gal80$^{ts}$-repressed 5-HT$_{2A}$R condition (blue), and at restrictive 28°C (right) for both control (green) and 5-HT$_{2A}$R OE condition (red). Two-way ANOVA with Tukey's multiple comparison tests shows there in no significant change in innervation volume in the Gal80$^{ts}$/repo control lines at 18°C ($n = 20$/condition, $p = 0.266$), or with the glial-targeted Gal80$^{ts}$-5-HT$_{2A}$R OE line at permissive 18°C ($n = 20$/condition, $p = 0.9529$), or in the Gal80$^{ts}$/repo control line at restrictive 28°C ($n = 20$/condition, $p = 0.870$). There is a significant decrease in the Or42a OSN innervation volume with glial-targeted Gal80$^{ts}$-5-HT$_{2A}$R OE at 28°C ($n = 20$/condition, $p = 1.00 \times 10^{-15}$). All the individual data points are shown with the mean ± SEM. Significance is indicated as $p < 0.0001$ (****) and $p > 0.05$ (not significant, ns). Source data can be found in S1 Data. dpe, days post-eclosion; OE, overexpression; OSN, olfactory sensory neuron; VM7, ventromedial 7.

driven by critical period sensory experience is a means to amplify remodeling signals that drive sensory experience-dependent synaptic glomeruli pruning in *Drosophila*. We find that serotonin production and 5-HT$_{2A}$ receptors specifically within glia are needed for experience-dependent synaptic glomeruli pruning. Serotonergic signaling is well known to regulate sensory input in higher order processing [24,25], and serotonin specifically controls experience-specific olfactory circuit plasticity in *Drosophila* [24,44,70], but this mechanism is mediated by the serotonergic CSDn neurons (not glia) in mature adults and operates on a much smaller scale than the experience-dependent synaptic glomeruli pruning within the juvenile critical period reported here. Thus, we have discovered a truly new requirement for glial serotonin signaling and 5-HT$_{2A}$ receptors that appears specific to the regulation of glial phagocytic synapse pruning in the early-life critical period. One mechanistic model to consider is the defined cellular "community effect" characterized during earlier developmental processes, in which regulated intercellular signaling coordinates orchestrated cell behaviors at important decision choice points [71,72]. A similar community effect mechanism may be operating in the *Drosophila* juvenile brain, with circuit-localized serotonergic signaling between glial cells necessary to coordinate the experience-dependent responses that initiate, enable, or maintain the glial infiltration phagocytic pruning of synaptic glomeruli within the temporally restricted developmental critical period (Fig 6).

We discover that glial serotonergic 5-HT$_{2A}$ receptors are essential for experience-dependent glial synaptic glomeruli pruning in the critical period. In neurons, 5-HT$_{2A}$ receptors are well known to regulate key learning/memory cascades in adult mice [33,73], and 5-HT$_{2A}$R autoregulation is also known to modulate serotonergic neuron circuit integration during development [74]. However, the glial 5-HT$_{2A}$R signaling discovered here in *Drosophila* appears entirely novel. We find experience-dependent up-regulation of glial Trhn 5-HT synthesis and 5-HT$_{2A}$R levels only in the juvenile critical period. In mice, neuronal 5-HT$_{2A}$R autocrine signaling is important for activity-dependent synaptic remodeling, which can be activated by a short-lived paracrine ligand (e.g., BDNF) to sustain and/or amplify the downstream signaling pathway [43,75,76]. Such retrograde neurotrophin signaling is a long-standing model for the control of autocrine feedback, but neurotrophins are not well conserved in *Drosophila* [77]. The glial serotonergic signaling shown here may similarly be an autocrine mechanism, or, alternatively, a glia-to-glia signaling mechanism responding to input experience to enable temporally restricted and circuit-localized glial pruning of synaptic glomeruli [44,45]. Either of these mechanisms is likely driven by dynamic changes in 5-HT$_{2A}$ receptor levels that directly coordinate sensory input experience, specifically during the critical period. It would be of interest to dissect the relationship between brain circuit synaptic remodeling and the regulated availability of 5-HT$_{2A}$ receptors in glial cells. This relationship could explain the tight restriction of experience-dependent synaptic pruning to critical periods [1,78]. Different glial subtypes may interact in this serotonin signaling mechanism, as we have shown in

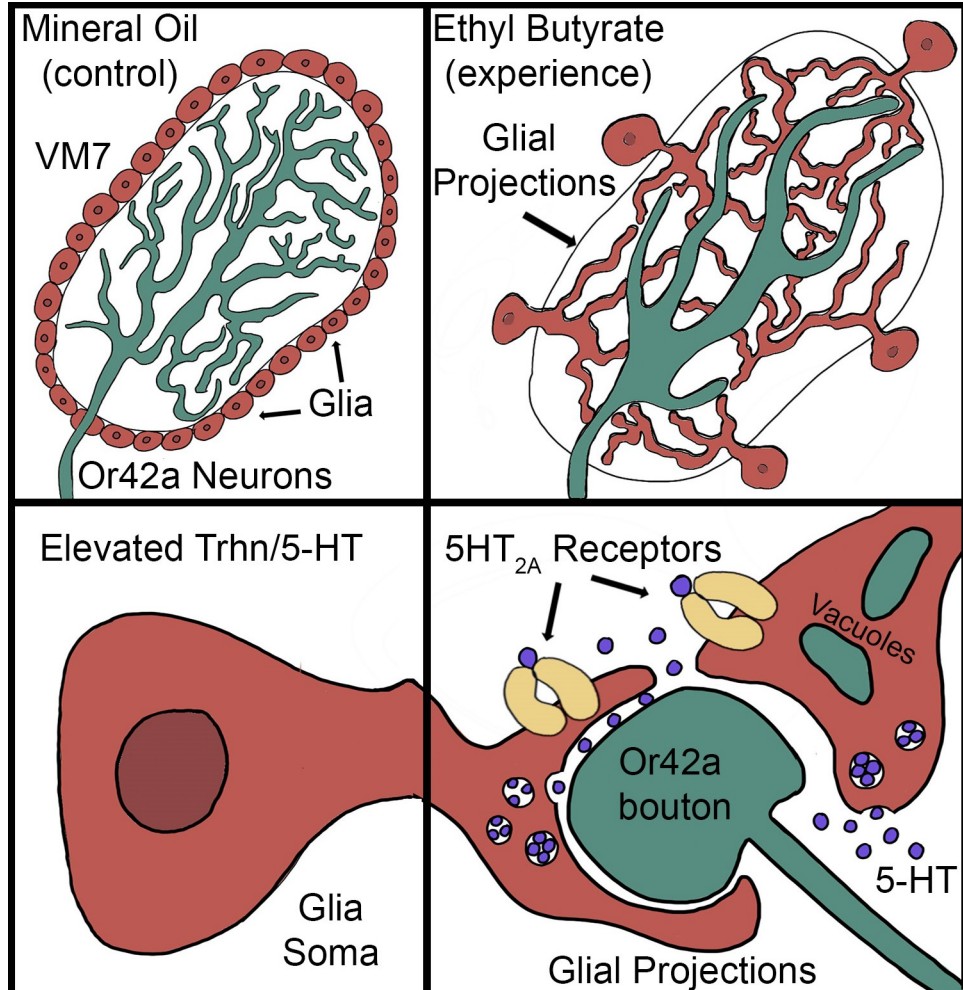

**Fig 6. Schematic of glial serotonergic signaling in critical period experience-dependent pruning of olfactory synaptic glomeruli.** Or42a OSN synaptic termini (green) densely innervate the VM7 glomerulus in normal conditions (top, left). Glia (red) predominantly reside outside the AL neuropil. With critical period exposure to EB odorant, glial processes (red) infiltrate the VM7 glomerulus (top, right). These projections engulf Or42a OSN termini and mediate the experience-dependent pruning of synapses for the optimization of circuit connectivity. With critical period experience, glial soma up-regulate Trhn to produce serotonin (5-HT; bottom, left). Glia release 5-HT (blue) in response to experience only during the critical period. Glial 5-HT$_{2A}$ receptors (yellow) are also up-regulated by critical period experience, with both glial Trhn serotonin production and 5-HT$_{2A}$ reception required for experience-dependent synaptic glomerulus pruning via bouton engulfment and phagocytosis by glia (bottom, right). AL, antennal lobe; EB, ethyl butyrate; OSN, olfactory sensory neuron; Trhn, tryptophan hydroxylase; VM7, ventromedial 7.

another case of glial phagocytosis circuit pruning in the *Drosophila* juvenile brain [46,47], and as occurs in other signaling contexts in rodents [79,80], but such a mechanism would be entirely novel within the critical period. Future studies will test whether the temporal-restriction of critical synaptic pruning is determined by glial-specific 5-HT$_{2A}$ receptor availability that provides a mechanistic capacity to sculpt brain circuits.

During the juvenile critical period, experience-dependent synapse pruning is strongly enhanced by glial 5-HT$_{2A}$ receptor OE, showing glial serotonin signaling is rate-limiting. This mechanism is glial-specific, as similar neuronal 5-HT$_{2A}$R OE has no effect on experience-dependent synaptic glomeruli pruning. In contrast, 5-HT$_7$R OE reverses metabotropic

glutamate receptor (mGluR)-mediated LTD in both wild-type mice and enhanced LTD in a neurodevelopmental disease model [81]. Pharmacologically, both 5-HT$_{2A}$R agonists and antagonists work in therapeutic measures to maintain baseline brain circuit function. For example, 5-HT$_{2A}$R antagonists (atypical antipsychotics) treatments work in only some schizophrenia patients, but the unresponsive cases to these antipsychotic medications often respond positively to a 5-HT$_{2A}$R agonist [73]. This rate-limiting signaling mechanism is also observed in PTSD patients. Similar to schizophrenia, PTSD-induced psychosis can respond to both 5-HT$_{2A}$R antagonists and selective serotonin reuptake inhibitors (SSRIs), but treatment-resistant PTSD cases can be responsive to both serotonergic signaling up-regulators (such as MDMA) [36,82] and 5-HT$_{2A}$R agonists (such as LSD and psilocybin) [39,40,83]. Thus, evidence for such 5-HT$_{2A}$R rate limitation dichotomy appears repeatedly, although it has been documented almost exclusive in the dysregulation of serotonergic signaling in neurological disorder states. In contrast, the rate-limiting function of glial 5-HT$_{2A}$Rs in experience-dependent synaptic glomeruli pruning appears to be entirely novel (Fig 6). Insights into 5-HT$_{2A}$R mechanistic requirements during critical periods, and specifically within glia, may provide key insights into new treatment avenues for a range of neurological disorders. One mechanistic model to consider is that the 5-HT$_{2A}$R dichotomy may exist owing to a "switch-like" requirement to sustain appropriate intercellular serotonergic signaling. In the large number of 5-HT$_{2A}$R-implicated disorders (e.g., schizophrenia, PTSD, dementia) [79], this switch-like signaling mechanism could respond to 5-HT$_{2A}$R antagonist/agonist treatments to restore appropriate brain circuit function.

This novel 5-HT$_{2A}$R requirement in glia could also provide insights for new treatments in glia-implicated neurodevelopment disorders such as Fragile X syndrome (FXS), the most common heritable cause of both intellectual disability (ID) and ASD [22,35]. Moreover, impaired serotonin signaling is implicated in many other disease states with hyperactive glial phagocyte brain circuit pruning (e.g., schizophrenia, neurodegenerative conditions) [73,84], suggesting a possible role for glial serotonin signaling dysregulation. 5-HT$_{2A}$Rs mediate normal immune pathway signaling [85,86], and hallucinogens, antipsychotics, and antidepressants all act via 5-HT$_{2A}$R function [40,87,88]. However, the mechanisms remain poorly understood across model species, including the cell types involved [73,89]. The *Drosophila* glial serotonin signaling reported here could potentially address the anomaly of 5-HT$_{2A}$R antagonists stochastically acting to both correct and exacerbate patient psychological symptoms [84,88,90]. Importantly, we discover glial OE of Trhn to drive 5-HT signaling, as well as 5-HT$_{2A}$ receptor conditional OE only in adult glia, reinitiates experience-dependent synaptic glomeruli pruning at maturity, reopening the previously "closed" critical period in our *Drosophila* model. Similarly, 5-HT$_{2A}$R agonists (e.g., LSD, psilocin) increase cultured adult mouse neuron plasticity [33,39,89], although the focus in these studies has been exclusively on growth factor pathways (e.g., BDNF and downstream TrkB signaling) that induce growth [40,76,91]. In a mechanistic signaling loop, conditional adult glia 5-HT$_{2A}$R OE reinitiates experience-dependent serotonin signaling. In future studies, it will be vital to explore glial signaling mechanisms that mediate experience-dependent synaptic glomeruli pruning in the critical period and maturity, across brain circuits, and in our *Drosophila* FXS disease model. The use of serotonergic therapeutics to induce de novo brain circuit remodeling at maturity is a fascinating objective [35,40,92], and the work reported here suggests glial serotonin signaling may be an important, overlooked component of the mechanism controlling experience-dependent synaptic connectivity changes.

## Methods

### *Drosophila* genetics

All animals were maintained at 25°C in 60% humidity with a 12:12-hour light/dark cycle on standard *Drosophila* food. The transgenic Gal4 activator driver lines were the following: ubiquitous *UH1*-Gal4 (RRID: BDSC 55850) [93]; glial *repo*-Gal4 (RRID: BDSC 7415) [94]; pan-neuronal *elav*-Gal4 (RRID: BDSC 8765) [57]; OSN *Or42a*-Gal4 (RRID: BDSC 9969) [49]; serotonergic neuron (CSDn) *GMR60F02*-Gal4 (RRID: BDSC 48228) [95]; and tryptophan hydroxylase *Trhn*-Gal4 [96]. The transgenic UAS responder lines were the following: membrane markers UAS-*mCD8::GFP* (RRID: BDSC 5137) [97] and -*mCD8::RFP* (RRID: BDSC 32219), UAS-*orco* (RRID: 23145) [98], and serotonin pathway UAS-*Trhn* RNAi (RRID: BDSC 25842) [58], UAS-*Trhn*$^{OE}$ (RRID: BDSC 27638) [99], UAS-*5-HT$_{2A}$* RNAi (RRID: BDSC 31882) [63], and UAS-*5-HT$_{2A}$*$^{OE}$ (RRID: BDSC 4830) [63]. *Or42a*-Gal4 was used to drive *UAS-orco* in *orco$^2$* null mutants (RRID: BDSC 23130) [98]. The temperature-sensitive (ts) Gal80 repressor was α-*tubulin-Gal80$^{ts}$* (RRID: BDSC 86328) [100]. The promoter fusion line was *Or42a-mCD8:GFP* [51]. The control lines were the genetic background *w$^{1118}$* (RRID: BDSC 3605), and the transgenic control lines (1) *w$^{1118}$; Or42a-mCD8:GFP/+; repo*-Gal4/+, (2) *w$^{1118}$; Or42a-mCD8:GFP/+; UAS-Trhn* RNAi/+, (3) *w$^{1118}$; Or42a-mCD8:GFP/+; UAS-5-HT$_{2A}$* RNAi/+, *w$^{1118}$; Or42a-mCD8:GFP/+; UAS-5-HT$_{2A}$*$^{OE}$/+, and (4) *w$^{1118}$; Or42a-mCD8::GFP/α-tubulin-Gal80$^{ts}$; repo*-Gal4/+. Animals of both sexes were used in all studies.

### Odorant exposure

Odor exposure treatments were done as previously described [44,45,101]. Briefly, developmentally staged animals were sorted as dark pupae into separate vials based on age, sex, and genotype. A fine wire stainless steel mesh was secured with taped Parafilm over the top of the vial. The vials were placed in an airtight 3,700 ml Glasslock container with 1 ml mineral oil vehicle (100%; Sigma-Aldrich) or 15% to 25% EB odorant (Sigma-Aldrich; % v/v EB in mineral oil) in a 1.5-ml microcentrifuge tube centered in the exposure chamber. The chambers were placed in temperature-controlled incubators (25°C) on a 12-hour light/dark cycle. All eclosed juvenile flies were rapidly transferred to clean vials in clean exposure chambers with freshly made odorants (as above), 4 hours after placing the vials into the chambers. The animals were kept in the odor exposure chambers in incubators for a further 20 hours (24 hours total) and then immediately processed for immunocytochemistry [101].

### Conditional transgenics

For all temperature-sensitive Gal80 (Gal80$^{ts}$) experiments, developmentally staged animals were sorted as dark pupae into separate vials based on age, sex, and genotype. Animals were reared for 14 days at 18°C (permissive temperature) and then transferred to the experimental temperature (maintained 18°C or 28°C restrictive temperature) for odorant exposure. As above, animals were exposed in vials with a fine wire stainless steel mesh top in airtight Glasslock containers to either 1 ml vehicle control only (100% mineral oil; Sigma-Aldrich) or 25% EB odorant (Sigma-Aldrich; % v/v EB in mineral oil). The animals were then maintained in the odor exposure chambers in temperature-controlled incubators for 24 hours before being immediately processed for immunocytochemistry.

### Immunocytochemistry imaging

Developmentally staged animals were anesthetized in 70% ethanol for 1 to 2 minutes and the brains dissected using sharpened forceps (Dumont #5) in 1x phosphate buffered saline (PBS;

Invitrogen). Brains were fixed for 30 minutes at room temperature (RT) in 4% paraformaldehyde (PFA; EMS 15714) in 4% sucrose PBS. Fixed brains were washed 3× with PBS and then blocked for 1.5 hours at RT or overnight (12 to 16 hours in 4°C) with 1% BSA (Sigma-Aldrich) in 0.2% Triton X-100 in PBS (PBS-T; Fisher Chemical). The brains were incubated with primary antibodies diluted in 0.2% BSA in PBS-T at 4°C overnight. The primary antibodies used were the following: chicken anti-GFP (Abcam, 13970; 1:1,000), rat anti-RFP (Chromotek, 5F8; 1:1,000), rat anti-DNEX-8 (Developmental Studies Hybridoma Bank (DSHB); 1:50), rabbit anti-5-HT (Immunostar; 1:1,000), rabbit anti-5HT$_{2A}$ receptor (Abcam, ab140524, 1:100), and mouse anti-Repo (DSHB, 8D12; 1:100). The brains were washed 3× for 20 minutes each with PBS-T and then incubated overnight with fluorescently conjugated secondary antibodies. The secondary antibodies used were the following: AlexaFluor-488 goat anti-rabbit, AlexFluor-488 goat anti-chicken, AlexaFluor-546 goat anti-rat, and AlexaFluor-546 donkey anti-rat (all used at 1:250). The brains were washed in PBS-T 3× for 20 minutes each, followed by PBS and dH$_2$O 1× for 20 minutes. Brains were mounted onto glass slides (75 × 25 mm, 0.9 to 1.06 mm; Corning) with a glass coverslip (No. 1.5H, Carl Zeiss) in Fluoromount-G (EMS 17984–25). Double-sided adhesive tape (Scotch) was used to raise coverslips over the brains, with clear nail polish (Sally Hansen) to seal coverslips. Images were collected on a 510 META laser-scanning confocal microscope (Carl Zeiss) with a 63× oil-immersion objective. Images were collected at 1,024 × 1,024 resolution with a Z-slice thickness of 0.75 μm [101]. The microscope and imaging settings were kept constant within every experiment (exact settings can be found in Protocol Exchange).

## Quantification measurements

All measurements were done blind to both genotype and experience conditions using the ImageJ Blind Analysis Tool plug-in. To quantify 5-HT, 5HT$_{2A}$R, and the tryptophan hydroxylase *Trhn*-Gal4 driving UAS-*mCD8*::*RFP* membrane marker intensity values, the weighted sum of all pixels was used, which adds together all the pixels in each slice at each position. Brightness values <50 were dropped to account for imaging background. The ImageJ JACoP analysis plug-in was used for colocalization tests. Pearson's coefficients were assayed on Z-stacks, with thresholds kept constant in blind comparisons. For Or42a innervation volume measurements, an ROI was defined for the borders of the VM7 glomerulus and innervation volume was quantified by lasso perimeter measurements from the sum slices Z-projection using the following equation: [*volume (μm$^3$) = area (μm$^2$) x slice thickness x total number of slices*]. Data from all the combined biological replicates were maintained as a raw measurement point spread, with blinded quantification to ensure normality across all trials [101].

## Statistical analyses

All statistical analyses were performed with Prism software (GraphPad version 9). All analyses were done using $N$ = number of synaptic glomeruli, unless otherwise stated. All groups that met the criteria for parametric statistics were analyzed with unpaired two-tailed $t$ tests. For data comparing $\geq 2$ genotypes, a two-way ANOVA was used with odorant exposure and genotype as independent variables, followed by Sidak's multiple-comparisons tests to compare the oil odorant vehicle and EB-exposure conditions within each genotype. Comparisons between 2 or more genotypes were analyzed by two-way ANOVA tests with a 5% alpha significance level. Data are presented in the figures as all the individual data points and the mean ± SEM. Significance in figures is indicated as $p < 0.05$ (*), $p < 0.01$ (**), $p < 0.001$ (***), and $p < 0.0001$ (****). Values of $p > 0.05$ are deemed not significant (ns). Exact significance $p$-values for each comparison are given in the figure legends.

## Supporting information

**S1 Fig. Critical period EB odorant experience via Or42a neurons specifically elevates VM7 serotonin signaling. (A)** UAS-*orco* driven with *Or42a*-Gal4 in *orco* null mutants (UAS-*orco*; *Or42a*-Gal4, *orco²*) enables the EB odorant response. Critical period exposure for 24 hours from 0–1 dpe to odorant oil vehicle (control, left) or 25% EB in oil (experience, right) with anti-serotonin (5-HT, green) and Or42a receptor-driven mCD8::GFP (Or42::GFP, magenta) membrane labeling for the VM7 innervation. **(B)** Quantification of 5-HT fluorescence intensity (left) and Or42a innervation volume (right) in VM7 glomeruli. Two-way ANOVA with Tukey's multiple comparisons show a significant increase in serotonin ($n$ = 12/condition, $p$ = 7.13 × 10⁻⁵) and a significant decrease in VM7 innervation ($n$ = 12/condition, $p$ = 4.78 × 10⁻¹⁰) with EB experience. Individual data points with mean ± SEM. Significance indicated as $p < 0.0001$ (****). **(C)** Genetic background $w^{1118}$ animals show no change in 5-HT (green) or CadN (magenta) in EB-independent DA1 glomeruli in response to 24-hour exposure from 0–1 dpe to odorant oil vehicle (control, left) or 25% EB in oil (experience, right). **(D)** Two-way ANOVA with Tukey's multiple comparisons show no significant change in 5-HT (green, $n$ = 10/condition, $p$ = 0.930) or CadN (magenta, $n$ = 10/condition, $p$ = 0.0.910) fluorescence intensity with EB experience. **(E)** Genetic background $w^{1118}$ animals show no change in 5-HT (green) or CadN (magenta) in the optic lobes in response to 24-hour exposure from 0–1 dpe to odorant oil vehicle (control, left) or 25% EB in oil (experience, right). **(F)** Two-way ANOVA with Tukey's multiple comparisons show no significant changes in 5-HT (green, $n$ = 10/condition, $p$ = 0.2981) or CadN (magenta, $n$ = 10/condition, $p$ = 0.0535) fluorescence intensity with EB experience. Individual data points shown with mean ± SEM. Source data can be found in S1 Data. CadN, N-Cadherin; dpe, days post-eclosion; EB, ethyl butyrate; VM7, ventromedial 7.
(TIFF)

**S2 Fig. Critical period glia express Trhn and serotonin (5-HT) with experience-dependent elevation. (A)** Critical period (0–1 dpe) VM7 synaptic glomeruli colabeled for glial nuclei (Repo, magenta) and a *Trhn*-Gal4-driven UAS-*mCD8::RFP* membrane marker (cyan). The comparison shows 24-hour exposure from 0–1 dpe to the oil odorant vehicle control (left) and 25% EB (right). **(B)** High magnification imaging following EB exposure with triple-labeling for serotonin (5-HT, yellow), glial nuclei (Repo, magenta), and *Trhn*-Gal4-driven UAS-*mCD8::RFP* membrane marker (blue). **(C)** Colocalization of 5-HT and Repo labeling (white) following 24-hour exposure from 0–1 dpe to the oil odorant vehicle control (left) and 25% EB (right). **(D)** Quantification of 5-HT and glial Repo colocalization with the 2 treatment conditions. An unpaired $t$ test comparison shows a significant increase in 5-HT-glia (Repo) colocalization ($n$ = 10 each, $p$ = 8.78 × 10⁻⁸) **(E)** Colocalization quantification of glial Repo and the *Trhn*-Gal4-driven UAS-*mCD8::RFP* membrane marker with the 2 treatment conditions. An unpaired $t$ test comparison shows significant increase in Trhn-glia (Repo) colocalization ($n$ = 10 each, $p$ = 0.0017). Source data can be found in S1 Data. dpe, days post-eclosion; EB, ethyl butyrate; Trhn, tryptophan hydroxylase; VM7, ventromedial 7.
(TIFF)

**S3 Fig. Cell-targeted *Trhn* RNAi in neurons and glia differential effects serotonin in the experience-dependent critical period. (A)** UAS-*Trhn* RNAi/+ transgenic control with critical period exposure for 24 hours from 0–1 dpe to 25% EB experience, with serotonin labeling (5-HT, green) in the VM7 glomerulus (left), and serotonergic neuron (CSDn) cell body (right). **(B)** *Trhn* RNAi driven by *elav*-Gal4 in neurons (*elav*>*Trhn* RNAi) under identical conditions. The EB experience-dependent serotonin up-regulation persists in the VM7

glomerulus (left), although serotonin is lost in the CSDn soma as expected (right). (**C**) *Trhn* RNAi driven by *repo*-Gal4 in glia (*repo>Trhn* RNAi) under identical conditions. The experience-dependent serotonin up-regulation is lost in the VM7 glomerulus (left), although serotonin is maintained within the CSDn soma (right). (**D**) Quantification of 5-HT fluorescence intensity in VM7. One-way ANOVA with Tukey's multiple comparison shows significant decrease in 5-HT in the *repo*-Gal4-driven UAS-*Trhn* RNAi compared to the UAS-*Trhn* RNAi/+ control ($n = 10$, $p = 9.00 \times 10^{-15}$) and *elav*-Gal4-driven UAS-*Trhn* RNAi ($n = 10$, $p = 5.72 \times 10^{-5}$). (**E**) Quantification of 5-HT fluorescence intensity in the CSDn neuronal cell body. One-way ANOVA with Tukey's multiple comparison shows a significant decrease in 5-HT in *elav*-Gal4 compared to *repo*-Gal4-driven UAS-*Trhn* RNAi ($n = 10$, $p = 8.00 \times 10^{-15}$). All individual data points are shown with the mean ± SEM. Significance is indicated as $p < 0.0001$ (\*\*\*\*), $p < 0.05$ (\*), and $p > 0.05$ (not significant, ns). Source data can be found in S1 Data. CSDn, contralaterally projecting, serotonin-immunoreactive deutocerebral neurons; dpe, days post-eclosion; EB, ethyl butyrate; RNAi, RNA interference; *Trhn*, *tryptophan hydroxylase*; VM7, ventromedial 7.
(TIFF)

**S4 Fig. 5HT$_{2A}$ receptor RNAi in neurons has no effect on experience-dependent pruning.** (**A**) *5-HT$_{2A}$R* RNAi control ($w^{1118}$; *Or42a-mCD8::GFP/+; UAS-5-HT$_{2A}$R RNAi/+*; <u>top</u>) and neuron-specific *5-HT$_{2A}$R* RNAi ($w^{1118}$; *Or42a-mCD8::GFP/elav*-Gal4; *UAS-5-HT$_{2A}$ RNAi/+*, <u>bottom</u>) following critical period exposure for 24 hours from 0–1 dpe to odorant oil vehicle (control, left) or 25% EB (experience, right). The *Or42a*-mCD8::GFP innervation of the VM7 glomerulus shown as a heat-map based on intensity (color LU scale; lower right panel). (**B**) Quantification of the Or42a neuron innervation 3-D volume. Two-way ANOVA with Tukey's multiple comparison shows significant pruning with EB experience ($n = 12$/condition, $p = 1.05 \times 10^{-12}$), which is not significantly different from the neuron-targeted *5-HT$_{2A}$R* RNAi condition ($n = 10$/condition, $p = 0.73$). Individual data points are shown with mean ± SEM. Significance indicated as $p > 0.05$ (not significant, ns). Source data can be found in S1 Data. dpe, days post-eclosion; EB, ethyl butyrate; RNAi, RNA interference; VM7, ventromedial 7.
(TIFF)

**S5 Fig. Experience-dependent up-regulation of 5HT2A receptors in the juvenile critical period but not in mature adults.** (**A**) Control ($w^{1118}$) juvenile critical period (0–1 dpe, top) or mature adult (7–8 dpe, bottom) AL staining of the 5HT$_{2A}$ receptors (anti-5HT$_{2A}$R, green) following exposure for 24 hours to oil (control, left) and 25% EB in oil (experience (EB), right). (**B**) Quantification of 5HT$_{2A}$R fluorescence intensity in both time periods and treatment conditions. Two-way ANOVA with Tukey's multiple comparison shows significant up-regulation in 5HT$_{2A}$R fluorescence intensity within the juvenile critical period with EB experience (green, left, $n = 10$/condition, $p = 1.12 \times 10^{-6}$), but no significant change in mature adults (red, right, $n = 10$/condition, $p = 0.123$). All individual data points are shown with mean ± SEM. Significance indicated as not significant (ns) at $p > 0.05$. Source data can be found in S1 Data. AL, antennal lobe; EB, ethyl butyrate; 5-HT$_{2A}$R, 5-HT2A receptor.
(TIFF)

**S6 Fig. Cell-targeted transgenic control of 5HT$_{2A}$R levels in neurons and glia during the experience-dependent critical period.** (**A**) Neuronal transgenic control (*elav*-Gal4/+, left) and neuronal knockdown of 5HT$_{2A}$ receptors (*elav*-Gal4/+, UAS-*5HT$_{2A}$* RNAi/+, right) with 5HT$_{2A}$ receptor labeling (green) following critical period exposure for 24 hours from 0–1 dpe to odorant vehicle (control, oil) or 25% odorant (experience, EB). (**B**) The same labeling with glial *repo*-Gal4 control (*repo*-Gal4/+, left), glial knockdown of 5HT$_{2A}$ receptors (UAS-*5-HT$_{2A}$*

RNAi/ *repo*-Gal4, middle), and glial 5HT$_{2A}$ OE (UAS-*5-HT$_{2A}$* OE/ *repo*-Gal4, right). (**C**) Quantification of 5HT$_{2A}$R fluorescence intensity in the *repo*-Gal4 control (green), *elav*-Gal4 control (blue), *repo*-Gal4 *5HT$_{2A}$* RNAi (orange), *elav*-Gal4 *5HT$_{2A}$* RNAi (yellow, second from the right), and 5HT glial *5HT$_{2A}$* OE (red). Two-way ANOVA with Tukey's multiple comparison shows EB experience-dependent increase in 5HT$_{2A}$ receptor levels in both transgenic controls ($n = 10$ each, *repo*-Gal4; $p = 5.38 \times 10^{-10}$, *elav*-Gal4; $p = 0.0205$), a significant decrease in 5HT$_{2A}$ receptor levels in both RNAi conditions ($n = 10$ each, *repo*-Gal4; $p = 4.63 \times 10^{-10}$, *elav*-Gal4; $p = 1.05 \times 10^{-9}$), and a significant elevated EB response with glial *5HT$_{2A}$R* OE ($n = 10$/ condition, $p = 2.36 \times 10^{-9}$). All individual data points are shown with the mean ± SEM. Significance is indicated as $p < 0.0001$ (****), $p < 0.05$ (*), and $p > 0.05$ (not significant, ns). Source data can be found in S1 Data. dpe, days post-eclosion; EB, ethyl butyrate; OE, overexpression; RNAi, RNA interference.
(TIFF)

**S7 Fig. Experience-dependent 5-HT up-regulation with 5HT$_{2A}$ receptor OE-induced synaptic pruning in mature adults.** (**A**) The conditional glial *5HT$_{2A}$R* OE line ($w^{1118}$; *tubulin-Gal80$^{ts}$*/*Or42a::GFP*; UAS-*5HT$_{2A}$R* OE/ *repo*-Gal4) in the permissive 18˚C Gal80$^{ts}$-repressed condition (top) and restrictive 28˚C *5HT$_{2A}$R* OE condition (bottom). VM7 colabeling for serotonin (5-HT, green) and Or42a innervation (Or42a::GFP, red) following mature adult exposure for 24 hours to either odorant vehicle (oil, left) or 25% odorant (EB, right). (**B**) Quantification of 5-HT fluorescence intensity in the permissive 18˚C Gal80$^{ts}$-repressed (blue) and restrictive 28˚C *5-HT$_{2A}$R* OE (red) conditions, in the oil control and EB exposure. Two-way ANOVA with Tukey's multiple comparison tests show no significant change in 5-HT fluorescence intensity in the 18˚C control ($n = 10$ each, $p = 0.789$), but a significant increase with glial-targeted *5HT$_{2A}$R* OE at 28˚C ($n = 10$ each, $p = 6.81 \times 10^{-8}$). Individual data points are shown with mean ± SEM. Significance is indicated as $p < 0.0001$ (****) and $p > 0.05$ (not significant, ns). Source data can be found in S1 Data. EB, ethyl butyrate; OE, overexpression; VM7, ventromedial 7.
(TIFF)

**S8 Fig. Conditional Trhn OE in adult glia induces EB experience-dependent 5-HT up-regulation and pruning.** (**A**) The conditional glial *Trhn* OE line ($w^{1118}$; *tubulin-Gal80$^{ts}$*/UAS-*Trhn* OE; *repo*-Gal4/UAS-*Or42a::GFP*) in the permissive 18˚C Gal80$^{ts}$-repressed condition (top) and 28˚C *Trhn* glial OE condition (bottom). VM7 colabeling for serotonin (5-HT, green) and Or42a innervation (Or42a::GFP, red) following mature adult exposure for 24 hours to either odorant vehicle (oil, left) or 25% odorant (EB, right). (**B**) Quantification of synaptic glomeruli innervation volumes in 18˚C control (blue) and 28˚C *Trhn* OE (red) conditions. Two-way ANOVA with Tukey's multiple comparison tests show no significant change in innervation volume at 18˚C ($n = 10$ each; $p = 0.985$), but a significant decrease at 28˚C with glial *Trhn* OE ($n = 10$ each, $p = 3.53 \times 10^{-10}$). (**C**) Quantification of 5-HT fluorescence intensities in 18˚C control (blue) and 28˚C *Trhn* OE (red) conditions. Two-way ANOVA with Tukey's multiple comparison tests show no significant change in 5-HT at 18˚C ($n = 10$ each, $p = 0.740$), but a significant increase at 28˚C with glial *Trhn OE* ($n = 10$ each, $p = 0.0412$). Individual data points are shown with mean ± SEM. Significance is indicated as $p < 0.0001$ (****), $p < 0.05$ (*), and $p > 0.05$ (not significant, ns). Source data can be found in S1 Data. EB, ethyl butyrate; OE, overexpression; Trhn, tryptophan hydroxylase; VM7, ventromedial 7.
(TIFF)

**S1 Data. Raw data values collected for all figures (Figs 2–5, S1–S8 Figs).** Each figure has a tab labels (i.e., "Main Fig 2") and is organized by panel label (A, B, C, etc.) in each tab.

Statistical analyses are included in S1 Data file.
(XLSX)

## Acknowledgments

We thank Zhichun Lin for contributions to *Drosophila* husbandry and genetic crosses. We thank Emma Rushton for contributions to the *Drosophila* genetics. We thank Dominic Vita for contributions to glial imaging. We thank other Broadie Lab members for key insights and constant discussions. We are grateful to the Bloomington *Drosophila* Stock Center and Vienna *Drosophila* Resource Center for essential lines, and the Developmental Studies Hybridoma Bank for essential antibodies.

## Author Contributions

**Conceptualization:** Vanessa Kay Miller.

**Data curation:** Vanessa Kay Miller.

**Formal analysis:** Vanessa Kay Miller.

**Funding acquisition:** Kendal Broadie.

**Investigation:** Vanessa Kay Miller.

**Methodology:** Vanessa Kay Miller.

**Project administration:** Vanessa Kay Miller.

**Resources:** Kendal Broadie.

**Supervision:** Kendal Broadie.

**Writing – original draft:** Vanessa Kay Miller.

**Writing – review & editing:** Kendal Broadie.

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
