## [Editor Report · Decision Letter 0]

29 Apr 2024

Dear Dr Broadie, 

Thank you for submitting your manuscript entitled "Experience-Dependent Glial Serotonin Self-Signaling Sculpts Brain Circuitry" for consideration as a Research Article by PLOS Biology.

Your manuscript has now been evaluated by the PLOS Biology editorial staff as well as by an academic editor with relevant expertise and I am writing to let you know that we would like to send your submission out for external peer review.

Once your full submission is complete, your paper will undergo a series of checks in preparation for peer review. After your manuscript has passed the checks it will be sent out for review. To provide the metadata for your submission, please Login to Editorial Manager (https://www.editorialmanager.com/pbiology) within two working days, i.e. by May 01 2024 11:59PM.

Kind regards,

Christian

Christian Schnell, PhD

Senior Editor

PLOS Biology

cschnell@plos.org

---

## [Decision Letter · Decision Letter 1]

6 Jun 2024

Dear Dr Broadie,

Thank you for your patience while your manuscript "Experience-Dependent Glial Serotonin Self-Signaling Sculpts Brain Circuitry" was peer-reviewed at PLOS Biology. It has now been evaluated by the PLOS Biology editors, an Academic Editor with relevant expertise, and by several independent reviewers. 

In light of the reviews, which you will find at the end of this email, we would like to invite you to revise the work to thoroughly address the reviewers' reports.

As you will see below, the reviewers are quite positive about your study but still raise a number of concerns that need to be addressed experimentally to strengthen the conclusions provided here.

Given the extent of revision needed, we cannot make a decision about publication until we have seen the revised manuscript and your response to the reviewers' comments. Your revised manuscript is likely to be sent for further evaluation by all or a subset of the reviewers.

**IMPORTANT - SUBMITTING YOUR REVISION**

*Re-submission Checklist*

*Published Peer Review*

*PLOS Data Policy*

*Blot and Gel Data Policy*

Sincerely,

Christian

Christian Schnell, PhD

Senior Editor

PLOS Biology

cschnell@plos.org

REVIEWS:

Reviewer #1: This study seeks to determine if serotonin signaling impacts a critical period in the olfactory system of Drosophila that depends upon glial cells. This group recently demonstrated that glia are responsible for a reduction in glomerular volume that is triggered by exposure to a high concentration of the cognate odorant for that glomerulus. In this study, they build upon this work to study the role of serotonin released by glia to act upon a glia expressed serotonin receptor. Unfortunately, there are several aspects of the data provided that raise concern which include quality of images and too little validation demonstrating that the genes targeted by RNAi are expressed by glia. 

Major concerns.

-The claim that glial cells synthesize and release serotonin in the brain of Drosophila is a very novel and requires strong supportive evidence, however very little evidence is provided. There are low-resolution immunolabeling images in which there may be colocalization between Repo-Gal4 (supplementary figure 2) and serotonin labeling, but they are not convincing because they are so cropped that they lack perspective and do not show glial cell bodies at all (supplementary figure 3). The other evidence is indirect in nature in which a tryptophan hydroxylase RNAi construct expressed in a glial driver line (Repo-Gal4) prevents the plasticity from occurring. There is no direct evidence demonstrating that glial cells synthesize serotonin, such as serotonin labeling of glial cell bodies or fluorescent in situ hybridization or transcriptomic analysis showing expression of serotonin synthesis enzymes. Simply checking FlyCell Atlas (a free online single-cell RNAseq dataset), there is no evidence for co-expression of repo with tryptophan hydroxylase or aromatic amino acid decarboxylase. Some form of direct evidence that glia produce serotonin is needed.

-The justification for testing the role of the 5-HT2A receptor in glial pruning is that this serotonin receptor is expressed by mammalian glia. However, the 5-HT2A receptor in Drosophila is not a homolog of the mammalian 5-HT2A, nor are the glia cells necessarily of similar cell types across these disparate groups. There needs to be some evidence that glia in Drosophila express the 5-HT2A receptor, otherwise expressing an RNAi construct with no validation of efficacy or off-target effects is not sufficiently rigorous to make any conclusion. Similar to the point above, a quick check in FlyCell atlas shows no expression of the 5-HT2A by glia in the brain of Drosophila, so there is a risk that effects caused by expressing could be artifactual without direct evidence that the 5-HT2A receptor is expressed by glia. 

-The increase in serotonin immunolabeling during the ethyl butyrate treatment as shown in figure 1C has two major flaws. First, the pattern of increased expression is clearly neuronal in nature with obvious branching compartments (Figure 1C and supplementary figure 1A) that are not consistent with reports of glial structure. This contradicts the suggestion that glial pruning is due to changes in glial, rather than neuronal, production of serotonin. The second issue is that the representative images are heavily cropped without showing serotonin labeling in the remainder of the brain. If this is a process involving the olfactory system, then there should be no change in labeling intensity in other brain regions such as the optic lobes. Fluorescence intensity measures must have internal controls (i.e. measures of fluorescence intensity from sources that are not impacted by a given manipulation) to ensure that any changes in fluorescence are not due to differences in confocal settings or other factors. 

Reviewer #2: Summary and Significance of Study

In this manuscript, Miller and Broadie report that autocrine serotonin signaling in glia regulates olfactory experience-dependent pruning in juvenile brains. Based on their studies, they also propose that expression of serotonin 5-HT2A autoreceptors is sufficient to re-open the critical developmental plasticity period in mature adults. This leads them to propose that glial serotonin autocrine signaling may yield intervention strategizes to alter experience-dependent plasticity remodeling in disease state.s 

This study advances the field in two ways. One, it reveals a role for glial autocrine serotonergic signaling in pruning. Two, it suggests how glial pruning is limited to developmental critical periods. This is a significant contribution that will be of interest to the fields of glia biology, developmental neurobiology, and neurological disease modeling. To make this work suitable for publication in PLOS Biology, the following points should be addressed:

Major Comments:

1. To support their conclusions, the authors need to demonstrate that EB does not change 5HT levels in non-VM7/Or42a innervated glomeruli, in at least one condition (wild type/Fig 1 or Or42a restricted/Fig S1). 

2. It is unclear how over-expression of the receptor alone is sufficient to prune in mature adults. The authors should show 5HT ligand staining with receptor OE and/or if TrpH over-expression phenocopies this adult reopening of plasticity.

3. The authors should show whether 5HT2AR levels change between juveniles (critical period) and mature adults to explain the critical period window per their model. 

4. Given prior implication of Draper in VM7 plasticity and altered expression with age, authors should examine Draper levels in juvenile EB and 5HT2AR OE conditions. 

Minor Comments: 

5. Some relevant references are missing and should be added e.g. PMID: 24270812, PMID: 33828296, PMID: 24361692, PMID: 37333889, PMID: 30150774, PMID: 25857335, PMID: 33156956

6. Line 42-44: The distinction made between "elimination…during experience dependent pruning" and "mediating pruning during experience dependent critical period" is unclear. Rephrase suggested.

7. A schematic in Figure 1 of the glomerulus and plasticity will be helpful to orient the reader.

8. Image quality needs improving. Some images/labels are hard to discern, for e.g.: 

-Fig 1A bottom right: is the text REPO? RFP? What is shown - glial projections are not visible or labeled by arrows. In Fig 1A top left, is the glomerulus outside the antennal lobe? -Fig 1C will be helped by also showing individual color panels. 

-Fig S2A top panels: what is to be inferred? There seem more glia cells in Fig S2A vs B, which makes the 5HT staining hard to infer. Schematic of which glia imaged will help. 

-Fig 2B: red and orange (repo vs elaV) colors are nearly identical

9. It is suggested that each panel is labeled for easy referencing for the reader. 

10. Overall, the text requires editing throughout for clarity and conciseness. For example, Line 148-149 is redundant with Line 159-160, since Fig 2B is a quantification of Fig. 2A. 

11. Repo:GAL4 expresses in multiple glia. The authors should clarify in text if they consider different glia to be signaling/receiving units or truly autocrine at single-cell level.

12. Line 161 needs statistical comparison in corresponding data in Fig 2B. 

Reviewer #3: 

Major points: 

1) The fact that neurons are not involved in this experience-dependent synaptic pruning is based only upon lack of effects of RNAi constructs of both Trhn-RNAi and 5-HT2A-RNAi (Figure 2 and Sup. Figure 3). It must be shown that these RNAi are both indeed working in the neurons by clearly reducing the level, obtained by immunolabeling or qRT-PCR, of 5-HT and receptor. 

2) The experience showing the re-open experience dependent synaptic pruning by adult receptors (Figure 4) should be bolstered by showing that this effect is indeed due to the concomitant upregulation of 5-HT. Also, the effect due to the glial expression of UAS-5-HT2A described should be abolished if UAS-Trhn-RNAi is co-expressed at the same time compared to a neutral UAS to have the same number of UAS present. 

3) The authors claim an autocrine mechanism, but this is not proven since they are using only repo-GAL4 which is expressed in all glia cell types. 5-HT could well be produced by one type of glia and used by another type that expressed the 5-HT2A receptor. Either the authors add results with a specific type of glia which show pruning defects with both Trhn-RNAi and 5-HT2A-RNAi, or they must remove the autocrine mechanism as proven. 

4) Why Figure S2 and S3 are not quantified? These quantifications must be done.

Minor points:

1) It will be very helpful for the readers to have a schematic model of the role of glia in this experience-dependent pruning. 

2) There is no mention of early work on Drosophila where glia cells were first recognized as phagocytes in axon pruning during mushroom body neuronal remodeling (Awasaki et al., 2004 and Watts et al., 2004). This is even more surprising that the corresponding author wrote a dispatch in the same issue of Current Biology to highlight these two seminal papers. 

3) line 367: Bloomington stock (25842) official name is Trhn RNAi and not TRH RNAi. This must be changed in all the manuscript, figures included. 

4) serotoninergic appears several times in the manuscript instead of serotonergic. 

6) Specify the complete genotype used in Figure 1 and Figure S2 figure legends.

7) It will be helpful to add UAS-Trhn RNAi in all appropriate panels in Figure 2.

8) The genotypes are unclear in the Figure 4 panels.

---

## [Editor Report · Decision Letter 2]

27 Aug 2024

Dear Dr Broadie,

Thank you for your patience while we considered your revised manuscript "Experience-Dependent Glial Serotonin Self-Signaling Sculpts Brain Circuitry" for publication as a Research Article at PLOS Biology. This revised version of your manuscript has been evaluated by the PLOS Biology editors and the Academic Editor.

Based on our Academic Editor's assessment of your revision, we are likely to accept this manuscript for publication, provided you satisfactorily address the following data and other policy-related requests:

* We would like to suggest a different title to improve accessibility and readability: 

Experience-dependent serotonergic signaling in glia regulates synapse elimination and targeting

* We would also recommend to edit your abstract for readability. For example, "Brain circuit connectivity optimization based on initial environmental input..." would be better as "The optimization of brain circuit connectivity based on..."

* Please tone down the first half of the final sentence of your abstract or remove it entirely, as the results of your study do not have direct relevance to clinical applications.

* Please add the links to the funding agencies in the Financial Disclosure statement in the manuscript details.

* DATA POLICY:

Regardless of the method selected, please ensure that you provide the individual numerical values that underlie the summary data displayed in the following figure panels as they are essential for readers to assess your analysis and to reproduce it: 2DE, 3B, 4C, 5C, S1BDF, S2DE, S3DE, S4B, S5B, S6C, S7B, and S8BC

* CODE POLICY

We expect to receive your revised manuscript within two weeks. 

*Published Peer Review History*

*Press*

Sincerely,

Christian

Christian Schnell, PhD

Senior Editor

cschnell@plos.org

PLOS Biology

---

## [Editor Report · Decision Letter 3]

29 Aug 2024

Dear Kendal,

Thank you for the submission of your revised Research Article "Experience-dependent serotonergic signaling in glia regulates targeted synapse elimination" for publication in PLOS Biology. On behalf of my colleagues and the Academic Editor, Bing Ye, I am pleased to say that we can in principle accept your manuscript for publication, provided you address any remaining formatting and reporting issues. These will be detailed in an email you should receive within 2-3 business days from our colleagues in the journal operations team; no action is required from you until then. Please note that we will not be able to formally accept your manuscript and schedule it for publication until you have completed any requested changes.

When you attend to the requests from my colleagues to come, please also add a statement to the corresponding figure legends where the raw data can be found. For example: "Source data can be found in S1 Data".

PRESS

Sincerely, 

Christian

Christian Schnell, PhD

Senior Editor

PLOS Biology

cschnell@plos.org